materials science/inorganic chemistry/physical chemistry

mixed matrix membrane, p84 co-polyimide, micro-mesoporous carbon, zeolite-templated carbon, three-dimensional graphene, gas separation

**Author for correspondence:**
Triyanda Gunawan
e-mail: triyanda@its.ac.id

This article has been edited by the Royal Society of Chemistry, including the commissioning, peer review process and editorial aspects up to the point of acceptance.

# The utilization of micro-mesoporous carbon-based filler in the P84 hollow fibre membrane for gas separation

Triyanda Gunawan[1], Nurul Widiastuti[1],
Hamzah Fansuri[1], Wan Norharyati Wan Salleh[2,3],
Ahmad Fauzi Ismail[2,3], Rijia Lin[4], Juliuz Motuzas[4]
and Simon Smart[4]

[1]Department of Chemistry, Faculty of Sciences and Data Analytics, Institut Teknologi Sepuluh Nopember, 60111 Sukolilo, Surabaya, Indonesia
[2]Advanced Membrane Technology Research Centre (AMTEC), and [3]Faculty of Petroleum and Renewable Energy Engineering, Universiti Teknologi Malaysia, 81310 Skudai, Johor Bahru, Malaysia
[4]School of Chemical Engineering, Faculty of Engineering, Architecture and Information Technology, The University of Queensland, St Lucia, Queensland 4072, Australia

TG, 0000-0003-0836-7715

This research involved carrying out a unique micro-mesoporous carbon particle incorporation into P84 co-polyimide membrane for improved gas separation performance. The carbon filler was prepared using a hard template method from zeolite and known as zeolite-templated carbon (ZTC). This research aims to study the loading amount of ZTC into P84 co-polyimide toward the gas separation performance. The ZTC was prepared using simple impregnation method of sucrose into hard template of zeolite Y. The SEM result showing a dispersed ZTC particle on the membrane surface and cross-section. The pore size distribution (PSD) of ZTC revealed that the particle consists of two characteristics of micro and mesoporous region. It was noted that with only 0.5 wt% of ZTC addition, the permeability was boosted up from 4.68 to 7.06 and from 8.95 to 13.15 barrer, for $CO_2$ and $H_2$ respectively when compared with the neat membrane. On the other hand, the optimum loading was at 1 wt%, where the membrane received thermal stability boost of 10% along with the 62.4 and 35% of selectivity boost of $CO_2/CH_4$ and $H_2/CH_4$, respectively. It was noted that the position of the filler on the membrane surface was significantly affecting the gas transport mechanism of the membrane. Overall, the results demonstrated that the addition of ZTC with proper filler position is a potential candidate to be applicable in the gas separation involving $CO_2$ and $H_2$.

# 1. Introduction

Separating gas via membrane technology is the most energy-efficient and has many opportunities for development, thus making this field of research grow fast at around 15% annually. A very exciting field, both scientifically and industrially, is gas separation using a polymeric membrane. In current industrial membrane gas separation technologies, both glassy and rubbery based polymers are widely used. The most popular rubbery polymers are ethylene oxide [1,2] and amide copolymer [3]. As well, polyimides [4–6], polysulfone [7,8] and polyamide imide [9] are the most referred glassy polymers. Polyimide-based polymers are some of the most used polymers for preparing gas separation membrane. Polyimides have good thermal properties ($T_g \sim 300°C$) and they can be easily prepared in various modules such as flat, tubular support and hollow fibre. Furthermore, they are affordable and provide good mechanical properties. In addition, these types of polymers are good for preparing carbon membranes [10–12].

Polymeric membrane for gas separation has a lot of drawbacks, especially in the trade-off between permeability and selectivity [13,14]. To overcome those drawbacks, many researchers tried to fix that by blending polymers [15] and preparing composite membrane [12,16]. Composite membrane is polymer membrane incorporated with inorganic materials. It proved to be a most effective and easy way to improve the membrane performance [17,18]. The idea is to combine the processability of the polymeric membrane with the selective adsorption and diffusion properties of the inorganic molecular sieve. Increasing or limiting the diffusion of gases in the membrane should improve the permeability and selectivity. Nanoporous material such as silica [19,20], zeolites [21,22], metal-organic framework (MOF) [23], graphene [24–27] and carbon nanotube [28,29] have been reported to form composite membrane. However, there are still technical challenges to be met such as avoiding pinhole formation and incompatibility issues with a polymer precursor.

Several studies have been reported on the low loading fillers in the mixed matrix membrane preparation [30,31]. Recently, carbon-based materials such as graphene gain a lot of attention in mixed matrix membrane progress. Graphene is a well-known two-dimensional carbon material composed of a single layer of carbon atom with a potential of $CO_2$ adsorption [31]. Moreover, it has excellent thermal and mechanical stability, and it is very beneficial to be embedded into polymer matrix to improve its physical properties. However, the use of graphene as a filler has a drawback owing to the impermeable nature of graphene that leads to the permeability reduction at high loading. So far, the optimal loading composition of graphene in mixed matrix membrane was around 0.3–1 wt% respective to the polymer mass [31–33]. At low loading, the permeability is still improved owing to the disruption of chain packing of the polymer matrix [33]. Whereas at high loading, the permeability reduction was observed owing to impermeable nature of graphene by constructing a barrier effect toward the gas flow [34,35]. The impermeable nature was the result of graphene being two-dimensional material that experiences unfavourable surface loss owing to stacking. Thus, to fully use the entire surface of graphene, an open three-dimensional network of self-standing graphene might be a potential candidate as a membrane filler.

According to Nishihara *et al.* [36], zeolite-templated carbon (ZTC) does realize the ideal of fully using the entire surface of graphene owing to self-standing open three-dimensional network. The ZTC can reach up to the 3707 $m^2\,g^{-1}$ geometric surface area which is exceeding the value of graphene (2627 $m^2\,g^{-1}$) owing to contribution of edge planes. In this study, ZTC, a structurally unique form of three-dimensional graphene, was prepared by the impregnation of sucrose into zeolite-Y pores and used as membrane filler in hollow fibre BTDA-TDI/MDI (P84) co-polyimide membrane for gas separation. Our motive uses this material as a filler, owing to the benefits of material such as high surface area, high microporosity and ordered pore structure [36,37]. With such characteristics, it is predicted that the use of ZTC into polymer matrix would improve the gas permeability owing to ordered free main path (ordered pore) and gas selectivity owing to high microporosity. As a result, making this material fits the required properties for the gas separation membrane. Previously, ZTC has decent $CO_2$ and $H_2$ adsorption capacity indicating good affinity towards the respective gas [37,38]. Thus, the gas flow behaviour needs to be controlled so that the adsorption of the stated gas can be avoided to improve the selectivity value. This research aims to improve the performance of P84 hollow fibre membrane for gas separation by using ZTC as a filler. Moreover, the effect of the filler position on the membrane surface was examined as well. The preparation and fabrication of ZTC/polymeric membrane composite was conducted at a series of ZTC/P84 composite membranes loaded with ZTC contents (0–1.5 wt%) by the dry/wet spin method. Detailed characterization of morphology (SEM), thermal stability (TGA), crystalline structure (XRD), topology (AFM) and functional group (FTIR) was conducted to understand the properties of nanocomposite polymeric membranes.

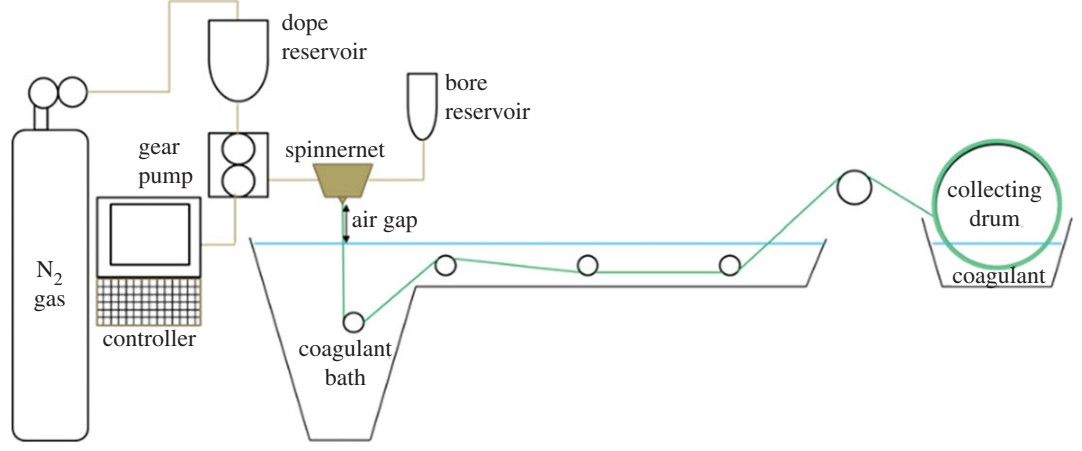

**Figure 1.** Schematic diagram of the dry/wet spin system.

# 2. Experimental section

## 2.1. Material

The ZTC was prepared using materials as follows; sodium aluminate (NaAlO$_2$, Sigma Aldrich) was used as a sodium source for zeolite-Y synthesis. Meanwhile, the silicate source was coming from sodium silicate (Na$_2$SiO$_3$, Sigma Aldrich). The additional sodium counter ion was provided by sodium hydroxide (99% NaOH, pellet, Sigma Aldrich). In addition, sucrose (98%, Fluka) was employed as a carbon precursor for ZTC filler. Hydrofluoric acid (48% HF, Sigma Aldrich) and hydrochloric acid (37% HCl, Merck) were used to remove the zeolite-template.

The raw materials for membrane preparation were P84 co-polyimide (BTDA-TDI/MDI, HP Polymer, Austria) as a polymer precursor and *N*-methyl-2-pyrrolidone (NMP, Merck) as the solvent. The P84 co-polyimide was dried at 80°C overnight to remove moisture prior to membrane preparation.

## 2.2. Procedure

### 2.2.1. Mixed-matrix membrane preparations

The ZTC filler preparation followed our previously reported method [37]. The hollow fibre membranes were fabricated using the dry/wet spin method illustrated in figure 1. The spinning parameters were inspired by the method reported by Favvas *et al.* and Choi *et al.* with some adjustment [39,40]. The ZTC filler (0–1.5 wt%) was first dispersed in NMP solvent using a sonicator (Qsonica, duration of 1 min, the amplitude of 70%, 10 s pulse on and off) for several times prior to the P84 addition. The P84 co-polyimide was then added bit by bit into the solvent, while mechanically stirred at 700 r.p.m. and temperature of 80°C. The dope solution that consists of P84/NMP/ZTC (20 : 80 : 0–1.5 w/w) and bore fluid of NMP/H$_2$O (70 : 30 w/v) were pumped simultaneously with a gear pump into a tube-in-orifice spinneret with the rate of 2.64 and 1.5 ml min$^{-1}$, respectively. The extruded fibres pass through a 400 and 800 µm inner and outer diameter spinneret, respectively. After passing the 5 cm of the air gap, the nascent fibres entering a tap water coagulation bath that was set at room temperature and oriented into a collecting drum with a speed of 4.5 m min$^{-1}$. The collected fibres were immersed in tap water to facilitate the solvent exchange overnight and followed by post-treatment of 2 h ethanol immersion at room temperature.

### 2.2.2. Sample characterization

X-ray diffractogram (XRD) was employed to confirm the structure formation of the ZTC and fabricated membrane. Fourier transform infrared (FTIR, Thermo Scientific Nicolet iS10) was employed to observe the alteration in the functional group of membrane and ZTC. The surface characteristic of ZTC was analysed using the N$_2$ adsorption–desorption isotherm at −195°C (Micromeritics, ASAP 2020). The sample morphology was observed using scanning electron microscope (Hitachi, TM 3000) and the results were analysed using ImageJ software. The potential used for SEM analysis was 15 kV and the

sample was coated with platinum. The thermal stability of the membrane was analysed using thermal gravimetric analyser (TGA, Brand-TA instrument TGA Q500). The surface roughness of the membranes was observed using atomic force microscopy (AFM, AIST-NT Inc., Novato, CA, USA) and the AFM images were analysed using AIST software.

### 2.2.3. Pure gas measurement

For the gas measurement, in all cases, five fibres (approx. 15 cm in length) were assembled in a laboratory-scale module. The fibres were potted in a Swagelok 3164B3, and the permeation performance was evaluated in a custom-made high-pressure gas permeation hollow fibre rig (1/4 in stainless steel (SS) 316 tubes), which connected directly to bubble meter. The single gas permeability was conducted at room temperature (approx. 25°C) and pressure of 4 bar. The gas volume was measured using the bubble flow meter. The measurement was in triplicate and the result presented here is the resulting average. The permeability was calculated using equation (2.1)

$$P_i = \left( \frac{Q \times l}{\Delta P \times A} \right) = \frac{Q.l}{n\pi D\Delta P}, \tag{2.1}$$

where $Q$ is the volumetric gas flow rate at standard temperature and pressure (cm$^3$ (STP) s$^{-1}$), $l$ is the membrane selective layer thickness (cm), $n$ is the number of fibres, $\Delta P$ is the different pressure between feed and system (cmHg), $A$ is the effective surface area of membrane (cm$^2$).

The ideal selectivity, $\alpha_{i/j}$, of the membrane was calculated using equation (2.2)

$$\alpha_{i/j} = \frac{P_i}{P_j}, \tag{2.2}$$

where $P_i$ is the permeability of gas $i$, and $P_j$ is the permeability of gas $j$.

For mixed-gas permeation, a method by Lin *et al.* was adopted [16]. The binary separation performance was conducted on two equimolar gas pairs of $CO_2/CH_4$ and $H_2/CH_4$.

# 3. Result and discussion

## 3.1. Filler preparation

Figure 2 illustrates the diffractogram pattern of ZTC showing a typical amorphous structure owing to the template removal process. Low structure replication of carbon to the zeolite-template structure was confirmed with the absence of the peak at approximately 6°C, which was also observed previously on the ZTC prepared with a similar impregnation method [37,41]. Furthermore, a broad weak peak that referred to the (002) mesophase graphite-like material was observed on the $2\theta$ of 20–25° [42,43]. Interestingly, wide-angle peaks at approximately 43°, which corresponds to the (101) of graphitic carbon, were not observed in this sample. This indicates that the ZTC material still replicates the structure of zeolite body and even possesses mesoporous characteristic. The reduced intensity in the composite diffractogram data indicates smaller particle transformation. Consequently, it would carry over to ZTC particle size characteristics and smaller particle size was expected.

The morphology of both zeolite-Y and ZTC is presented in figure 3. As can be seen in figure 3, the morphology of zeolite-Y shows hexagonal, diamond, rhombic and triangular crystal morphology. The smooth surface and sharp particle edge in zeolite Y indicate that preparation using gel method produced high crystallinity of zeolite. Similar morphology was also observed in the ZTC micrograph, which is in agreement with the XRD result. Moreover, the ZTC particle size was almost twice as small as the zeolite-Y particle. The particle size distribution of ZTC was also sharper than the zeolite. This indicates that the removal of zeolite-template not only produced smaller particle than the zeolite, but also produced more homogeneous particle size distribution. It is suggested that zeolite framework shrinking took place during the carbonization of the composite and more compact structure formed. This mechanism was similar to the mechanism of sintering process. Generally, smaller particle size was more preferred in fabricating the mixed matrix membrane, owing to more dispersed particle distribution on the membrane and avoiding the aggregate formation [7,16]. The TEM observation revealed that there was a thin carbon layer covering the ZTC particle and originated from the excess sucrose molecule and responsible in the absence of $2\theta$ approximately 6° peak on ZTC. This thin carbon layer was similar to the graphene structure reported by Geng *et al.* [44]. The inset image in figure 3*d*

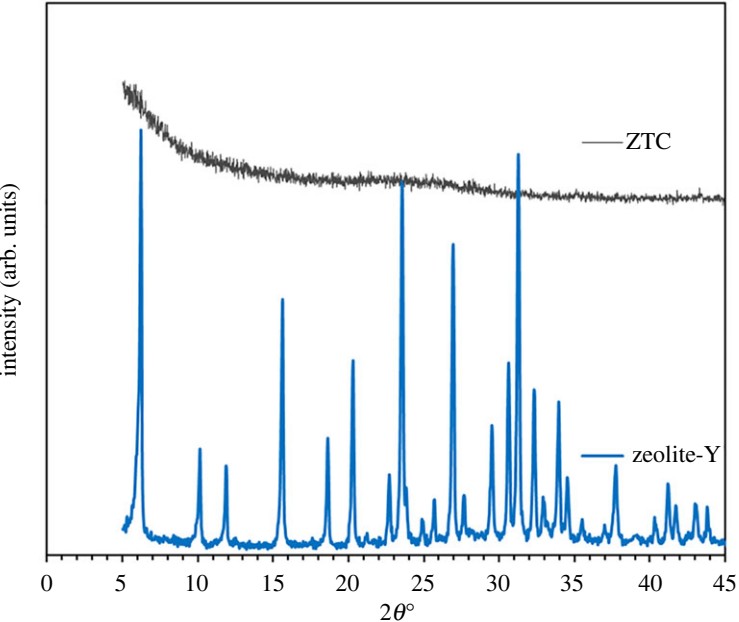

**Figure 2.** The diffractogram of zeolite Y, composite and ZTC.

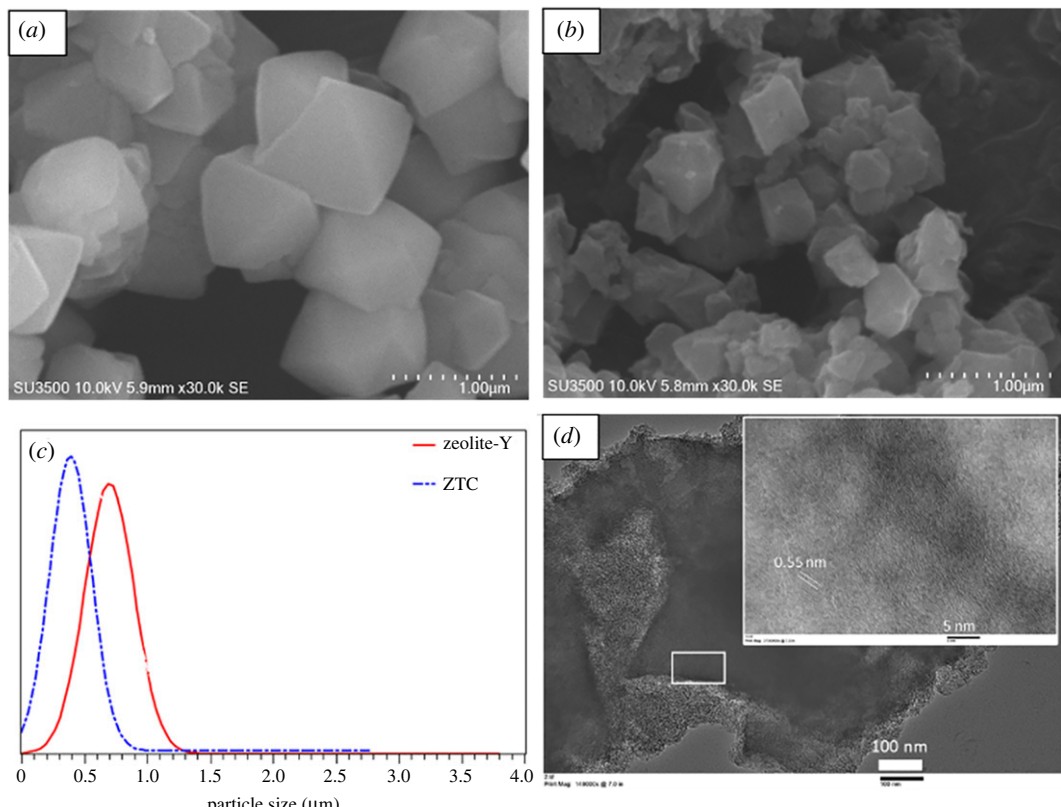

**Figure 3.** Zeolite-Y (*a*) and ZTC (*b*) SEM image, particle size distribution (*c*) and TEM image of ZTC (*d*), circle mark is the outside carbon layer.

shows that the main body of ZTC was consist of well ordered-interconnected pores. Overall, the SEM and TEM indicate that ZTC main body consisted of three-dimensional graphene network, while the outer layer was two-dimensional graphene sheet.

Figure 4 shows the $N_2$ adsorption–desorption isotherm graph and PSD of zeolite and ZTC. Both zeolite-Y and ZTC showed typical type I adsorption isotherm, which corresponds to the microporous

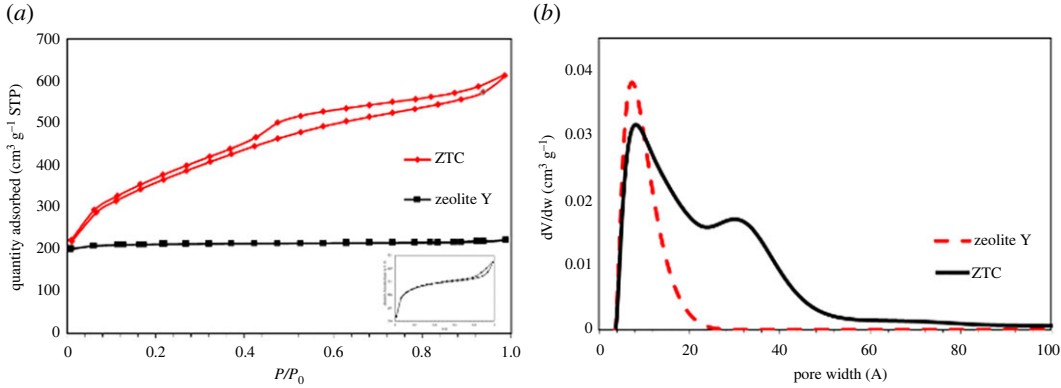

**Figure 4.** The isotherm (*a*) and PSD (*b*) of ZTC and zeolite-Y. Inset in (*a*) was refers to the zeolite-Y isotherm.

**Table 1.** Isotherm parameters of zeolite-Y and ZTC. $S_{BET}$, total surface area determined from Brunauer–Emmett–Teller; $S_{mic}$, total surface area of micropore, determined from *t*-plot; $S_{ext}$, total external surface area, determined from $S_{BET} - S_{mic}$; $V_{tot}$, total pore volume.

| sample | $S_{BET}$ (m$^2$ g$^{-1}$) | $S_{mic}$ (m$^2$ g$^{-1}$) | $S_{ext}$ (m$^2$ g$^{-1}$) | $V_{tot}$ (cc g$^{-1}$) | average pore size (Å) |
|---|---|---|---|---|---|
| zeolite Y | 678.48 | 620.0 | 35.77 | 0.34 | 8.61 ± 0.07[a] |
| ZTC | 1254.38 | 1051.72 | 202.66 | 0.95 | 15.5 ± 0.64[a] |
| | | | | | $\alpha = 9.23 \pm 0.10$[b] |
| | | | | | $\beta = 24.55 \pm 0.84$[c] |

[a]Total average pore size.
[b]Average pore size in $\alpha$ region.
[c]Average pore size in $\beta$ region.

material. However, the type H4 hysteresis was observed in the ZTC, which suggests the presence of mesopores and narrow slit pores [37,45]. The important parameters obtained from $N_2$ adsorption/ desorption are listed in table 1. The specific surface area ($S_{BET}$) of ZTC was almost double the zeolite-Y, reaching up to 1254.38 m$^2$ g$^{-1}$ when compared with the 678.48 m$^2$ g$^{-1}$ of zeolite-Y $S_{BET}$. High $S_{BET}$ of ZTC was also accompanied by the superior total pore volume that reached up to 0.95 cc g$^{-1}$ or 2.7 times bigger than the zeolite-Y pore volume. High surface area and pore volume of a filler were expected to improve the gas permeability owing to more accessible alternative pathway. The PSD was determined using SAIEUS software with 2D-NLDFT model [37]. Interestingly, the PSD of ZTC shows two types of pore region of microporous and mesoporous. The microporous region has pore diameter of 9.23 ± 0.10 Å, while the mesoporous region has 24.55 ± 0.84 Å average pore size. Our previous HRTEM study reported that the mesoporous region lies on the outer part of ZTC and has random pore structure, while the microporous was on the inner part of ZTC body with ordered and interconnected pore structure [37]. With such configuration, it is expected that the mesoporous would improve the gas permeability, while the microporous region would improve the selectivity.

## 3.2. Mixed-matrix membrane preparation

The alteration in functional groups in P84 co-polyimide after ZTC incorporation was studied using Fourier transform infrared (FTIR). Figure 5 shows the FTIR spectra of all prepared samples. First of all, the PR membrane was prior to ethanol post-treatment membrane. When compared with the other membrane spectra, it has a higher intensity at around 1650 cm$^{-1}$, coming from the excess NMP solvent. Meanwhile, all post-treated membranes have similar spectra patterns when compared with the P84 powder at this region. This indicates that all NMP were fully discarded from the membrane. It can be noticed that all the membranes showed typical polyimide peak at wavenumber of 720, 1360, 1715 and 1780 cm$^{-1}$. The peak at 720 cm$^{-1}$ corresponds to the C=O bond from P84 co-polyimide precursor. Band at 1350 cm$^{-1}$ corresponds to the C–N, while bands at 1715 and 1780 cm$^{-1}$ correspond

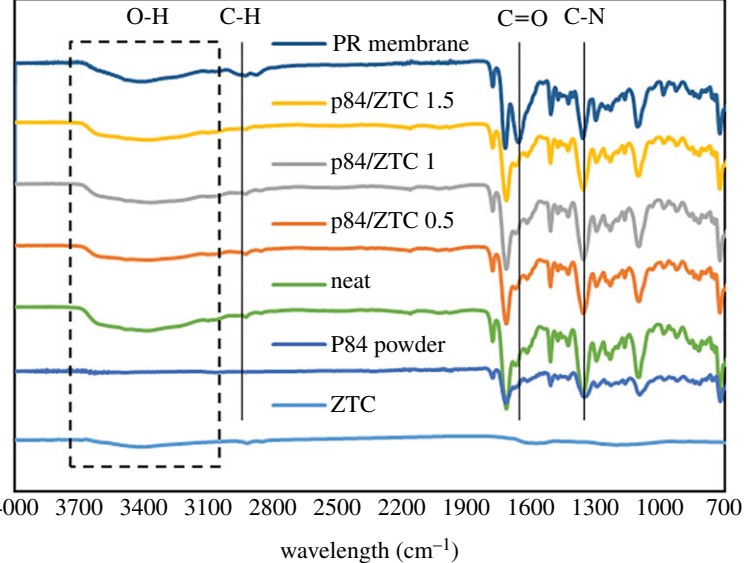

**Figure 5.** The FTIR spectra of all prepared samples.

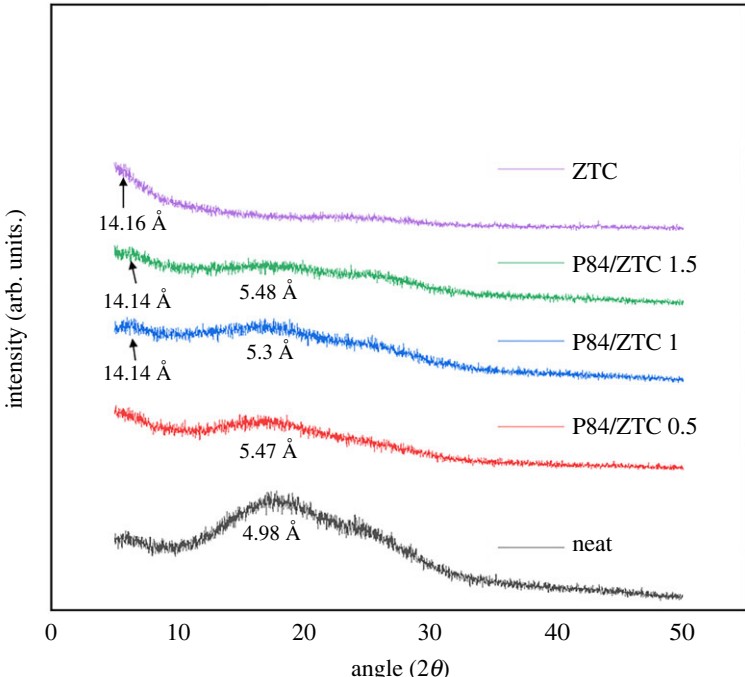

**Figure 6.** The XRD diffractogram pattern of prepared membrane.

to the C=O symmetric and asymmetric, respectively. The broad peak that appeared at around 3400 cm$^{-1}$ refers to the hydrogen bonding of trapped moisture [12,46], while the ZTC peak on the mixed matrix membrane overlapped with the P84 peak. Moreover, there was no new peak observed in the membrane with ZTC incorporation, indicating that the interaction between P84 co-polyimide and ZTC was physical interaction.

Figure 6 shows the diffractogram of all prepared membranes. Generally, the XRD pattern of polymer with large crystalline region reveals high intensity/sharp peaks, while low intensity/broader peaks confirm the amorphous region [47]. The diffractogram of all membranes showing typical amorphous structure. The broad peak at 2θ of 10–35° corresponds to the amorphous structure of P84 co-polyimide membrane, in agreement with previously reported work [46]. Since both ZTC and P84 co-polyimide were amorphous, it was hard to observe the difference in peak after the incorporation of ZTC. The only noticeable alteration was the reduction of intensity in the broad peak. This confirms

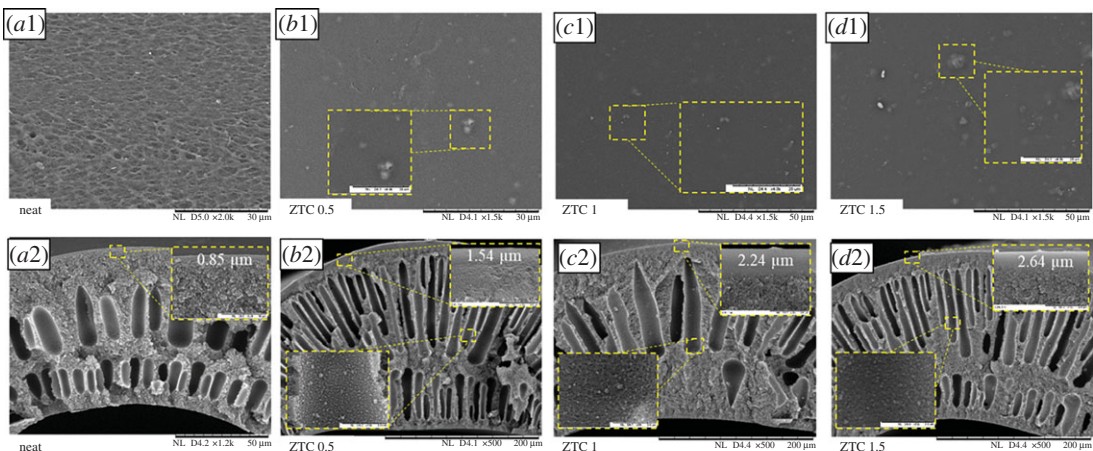

**Figure 7.** The surface (1) and cross-section (2) image of neat membrane (*a*), P84/ZTC 0.5 (*b*), P84/ZTC 1 (*c*) and P84/ZTC 1.5 (*d*).

that the primary semi-crystalline internal structure of P84 co-polyimide is being changed into a more rigid phase, owing to the adhesion force, throughout the addition of ZTC.

The membrane appearance was significantly different between the neat and ZTC filled membrane (electronic supplementary material, figure S1). The P84/ZTC 1 membrane showing a black colour which indicates the well-dispersed filler distribution, which was later confirmed with the SEM observation. The morphological structure of both the surface and cross-section are illustrated in figure 7. The ZTC particle distribution was homogeneous along the membrane surface even at low loading. Furthermore, since ZTC is a carbon-based material, the particle–filler interface showed no interfacial gap around the ZTC particle. Moreover, the ZTC particle on the membrane surface was mostly covered by the polymer body. This indicates that ZTC as carbon-based material was compatible with the polyimide. The aggregate formation on the membrane surface was observed at the loading of 1.5 wt%. In contrast to the membrane surface, the ZTC particle was distributed homogeneously on the membrane cross-section. There was almost no agglomeration of the ZTC particle, even at high loading. Mostly one particle of ZTC was covered by polymer cocoon. This could happen owing to good adhesion between ZTC and P84 co-polyimide during the hollow fibre fabrication. It is suggested that the ZTC particle in this region was mostly occupied on the top of the middle part of the dope solution during the spinning process. By then, the particle movement was limited in order to form agglomerate owing to being covered by the polymer. On the contrary, the particle on the surface of the membrane might come from the ZTC that occupied the bottom part of the dope solution. These particles were already agglomerated prior to the polymer addition during the dope solution preparation. This is commonly observed in the carbon-based filler in the mixed matrix membrane preparation at high loading [48–50]. The membrane surface was getting smoother after the addition of ZTC, which was in agreement with the XRD data. The polymer chain experienced the rearrangement after the introduction of filler owing to the adhesion force between filler and polymer body. This was indicated by the reduction of amorphous phase ($2\theta \sim 15°$) on the XRD pattern. Moreover, the presence of the concentric cavities in the membrane indicates that there is a strong interaction between polymer and filler [16]. The dense layer thickness gradually increased as the fillers loading amount improved. The increment in the dense layer was originated from the improvement of dope solution viscosity and thus, it improved permeability [51].

The surface topology is an important parameter to give a deeper understanding of the filler behaviour on the membrane surface. The observed surface roughness of membrane after ZTC addition was conducted by topology study using AFM as seen in figure 8. The data obtained from AFM observation was further analysed using AIST-NT SPM control software. In order to understand how well the filler attaches on the surface, the median level was applied to the image. This mode allowed to neglect the insignificant peak and valley so that the significant peaks from the filler were able to be measured precisely. As can be seen in the image, the roughness value (Ra) of the membrane was escalated as the ZTC filler was introduced. It keeps increasing as the filler loading is enlarged, which is pretty obvious owing to having a more extraneous fraction on the surface [52]. At ZTC 0.5 wt% incorporation, the peak length and height was 74.6 nm and 0.13 µm, respectively. This peak corresponded to the single ZTC particle on the membrane surface and indicated that the filler tends to

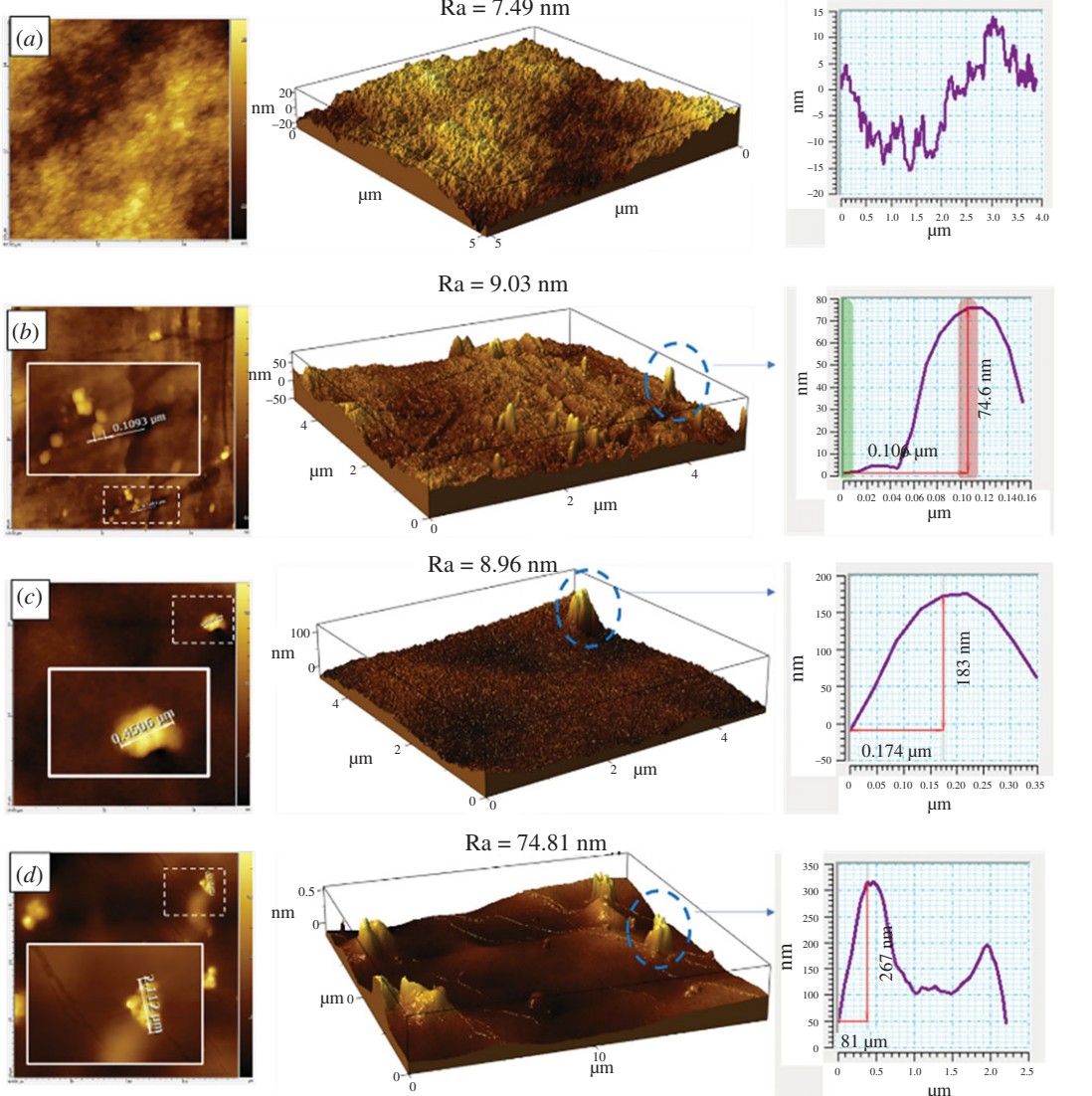

**Figure 8.** AFM image of (*a*) neat, (*b*) P84/ZCC 0.5, (*c*) P84/ZCC 1 and (*d*) P84/ZCC 1.5 in median level. Inset image corresponds to the original image as observed.

descend into the membrane surface, leaving just a tip on the surface. The ZTC particle began to ascend as more filler was added. At the addition of 1 wt% of ZTC, the observed peak shows the best filler exposure to the surface, reaching up to 183 nm or 46% of the particle body, and the length of the peak was 0.45 µm. It was in the range of one ZTC particle size, meaning no aggregate formation. At higher loading of 1.5 wt %, a bunch of stacked peaks were observed on the membrane surface. This indicates the aggregate formation of ZTC on the membrane surface. Based on the peaks length of 2.27 µm, it was assumed that there were seven particles of ZTC forming an aggregate and almost 70% of particle exposure to the surface. Since the ZTC particle shape was octahedral, an ideal filler exposure should be around 50%. In that manner, the filler would provide sufficient surface area to have contact with the permeate gas. Moreover, the polymer would provide enough support to hold the particle in place, when the membrane experienced pressure force during the separation process. While the appearance is far below or beyond that point, the filler contribution will be limited. When the ZTC appearance is far below 50%, the contact between the filler and the permeate gas will be limited, thus it would result in slight improvement toward gas separation performance. Meanwhile, if the appearance is far beyond 50%, only a few parts of the ZTC body are held by the polymer. This could make the filler peel off from the membrane when experiencing pressure during the separation process.

It is speculated that these filler positions were formed during the dope solution and membrane fabrication. At low filler loading, ZTC particle could be dispersed in the NMP solvent with ease.

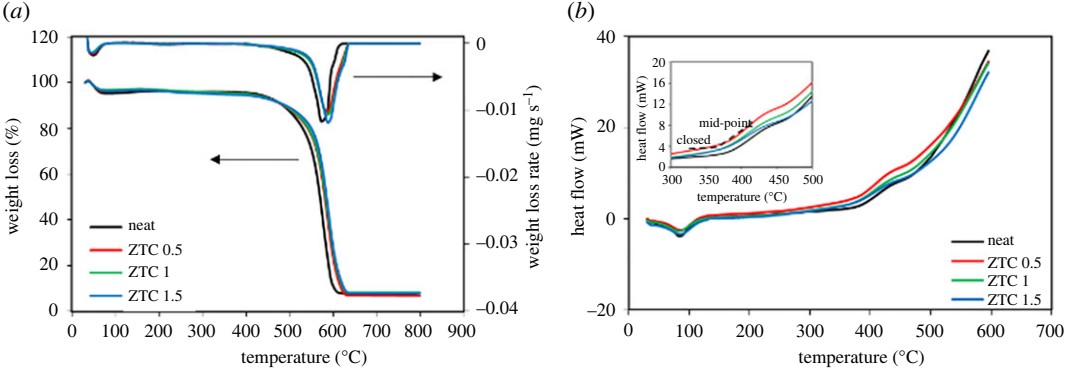

**Figure 9.** (*a*) TGA/DTA and (*b*) DSC curve of all prepared membranes.

**Table 2.** Thermal properties of all MMMs.

| sample | filler wt% | $T_g$ (°C) | $T_d$ (°C)[a] | resiowing (%)[b] |
|---|---|---|---|---|
| neat | 0 | 315 | 531.33 | 7.92 |
| ZTC 0.5 | 0.5 | 315 | 547 | 7.03 |
| ZTC 1 | 1 | 318 | 586.17 | 8.32 |
| ZTC 1.5 | 1.5 | 318 | 578.33 | 8.12 |

[a]$T_d$ is determined from the minimum value of DTA data.

[b]Resiowing is determined from the final mass of the sample at the end of TGA analysis, which is at 800°C.

However, at high loading, more insoluble ZTC tends to form agglomerate and occupied the bottom part of Duran flask owing to the increase of total weight. After the addition of P84, these aggregates could come out first during the dope solution pouring in the spinning reservoir owing to the gravitational force. Consequently, the filler tends to rise on the membrane surface, resulting in the position on the filler looking like the self-standing material. It is suggested to apply coupling agent on the filler prior to the dope solution formation to have better filler dispersion [47,53,54].

The thermal resistance study was conducted using TGA (figure 9). The degradation temperature of P84 membrane after the incorporation of ZTC particle were determined and tabulated in table 2. As can be seen in figure 9, the degradation of all the prepared membranes was in two steps. The first was the evaporation of excess water at 100°C, approximately 5 wt% mass loss in each membrane. While the second was the starting decomposition of P84 co-polyimide that was different for each sample. The pristine membrane began to decompose at temperature of 531.33°C, which was consistent with the previously reported data [39,40,55]. This indicates the high thermal stability of a P84 co-polyimide. The addition of ZTC particle gave an improvement toward the decomposition temperature of the P84 membrane, with notably the highest improvement at 1 wt% of ZTC into 586.17°C (10.32% improvement), followed by ZTC 1.5 and ZTC 0.5 with thermal stability improvement of 578.33 and 547°C, respectively. The resiowing produced after the introduction of ZTC particle into membrane was 7.03, 8.32 and 8.12% for 0.5, 1 and 1.5 wt% loading, respectively. Generally, inorganic filler plays as heat absorber during the heating process, resulting in improvement of thermal properties [7,56]. It is well known that as carbonaceous based material, ZTC possess high thermal stability, chemical stability and hydrophobicity that play an important role to improve the thermal stability of the membrane [36]. High thermal stability material means that this material could adsorb more heat before transporting it into the polymer surface [57]. While chemical stability keeps the filler and polymer in contact physically, the polymer structure is not changed after the filler filling. As the P84 is hydrophobic, having a hydrophobic filler is preferred since it has better adhesion owing to character similarity. This leads to the thermal stability improvement [58].

The glass transition temperature ($T_g$) of all the prepared membrane was determined by differential scanning calorimetry (DSC). Generally, filler particle would give a plasticization effect to the polymer membrane that would reduce the $T_g$ value [58]. The improved $T_g$ value (table 2) after filler

**Table 3.** The gas permeability of the studied MMMs in this study.

| membrane | permeability (barrer) | | | |
| --- | --- | --- | --- | --- |
| | $CH_4$ | $N_2$ | $CO_2$ | $H_2$ |
| uncoated PDMS | | | | |
| neat | $1.55 \pm 0.09$ | $1.73 \pm 0.03$ | $13.38 \pm 0.33$ | $22.72 \pm 0.93$ |
| P84/ZTC 0.5 | $1.94 \pm 0.03$ | $1.87 \pm 0.02$ | $14.98 \pm 0.62$ | $35.45 \pm 1.35$ |
| P84/ZTC 1 | $2.26 \pm 0.01$ | $2.18 \pm 0.02$ | $24.53 \pm 1.24$ | $47.44 \pm 2.69$ |
| P84/ZTC 1.5 | $2.30 \pm 0.05$ | $2.16 \pm 0.24$ | $17.55 \pm 1.07$ | $34.92 \pm 1.69$ |
| coated PDMS | | | | |
| neat | $0.21 \pm 0.00$ | $0.22 \pm 0.00$ | $4.68 \pm 0.07$ | $8.95 \pm 0.31$ |
| P84/ZTC 0.5 | $0.35 \pm 0.00$ | $0.36 \pm 0.01$ | $7.06 \pm 0.20$ | $13.15 \pm 1.61$ |
| P84/ZTC 1 | $0.55 \pm 0.01$ | $0.64 \pm 0.03$ | $19.57 \pm 0.76$ | $31.09 \pm 4.57$ |
| P84/ZTC 1.5 | $0.65 \pm 0.09$ | $0.66 \pm 0.01$ | $13.21 \pm 1.75$ | $29.10 \pm 2.34$ |

**Table 4.** The single gas selectivity of the studied MMMs in this study.

| membrane | selectivity | | | |
| --- | --- | --- | --- | --- |
| | $CO_2/CH_4$ | $CO_2/N_2$ | $H_2/N_2$ | $H_2/CH_4$ |
| uncoated PDMS | | | | |
| neat | $8.64 \pm 0.59$ | $7.73 \pm 0.17$ | $13.14 \pm 0.64$ | $14.66 \pm 0.76$ |
| P84/ZTC 0.5 | $7.72 \pm 0.40$ | $8.00 \pm 0.34$ | $18.92 \pm 0.75$ | $18.26 \pm 0.91$ |
| P84/ZTC 1 | $10.87 \pm 0.57$ | $11.28 \pm 0.62$ | $21.82 \pm 1.39$ | $21.02 \pm 1.22$ |
| P84/ZTC 1.5 | $7.65 \pm 0.59$ | $8.26 \pm 1.30$ | $16.32 \pm 1.48$ | $15.21 \pm 0.60$ |
| coated PDMS | | | | |
| neat | $21.89 \pm 0.28$ | $20.93 \pm 0.49$ | $39.98 \pm 1.44$ | $41.81 \pm 1.50$ |
| P84/ZTC 0.5 | $20.02 \pm 0.59$ | $19.61 \pm 0.30$ | $36.50 \pm 4.35$ | $37.29 \pm 4.80$ |
| P84/ZTC 1 | $35.54 \pm 1.71$ | $30.68 \pm 1.61$ | $48.86 \pm 5.57$ | $56.53 \pm 4.17$ |
| P84/ZTC 1.5 | $20.25 \pm 1.78$ | $19.99 \pm 2.68$ | $44.03 \pm 3.59$ | $45.09 \pm 6.99$ |

introduction means there is macromolecular chain rigidification that occurs in the polymer. This could affect the gas transport properties of the membrane, expecting an improvement in selectivity but decreasing permeability coefficients [59]. For all the membrane prepared here, the noticed improved $T_g$ value was at loading of 1 wt% ZTC filler. It indicates that the macromolecules chain in the polymer experiencing rigidification at loading amount. This result is in good agreement with the XRD data.

## 3.3. Single gas permeation

The separation performance of MMMs membrane was evaluated to determine the optimum loading of each filler. The permeation test was conducted on $CH_4$, $N_2$, $H_2$ and $CO_2$ gas, in that order at room temperature and 2 bar of feed pressure. Five fibres of 15 cm in length were potted in a Swagelok 3164B3 and sealed with epoxy ($2:1$ w/w resin : harderner) prior to the measurement. The single gas measurement was conducted at room temperature (approx. 25°C) and 2 bar feed pressure. The permeate side was connected to the bubble flow meter where the volumetric flow is acquired. The permeability and selectivity performance of the MMMs were listed in tables 3 and 4. The results obtained on all of MMMs after membrane potting were much higher than those reported in the literature. For the neat membrane, the $CO_2$ permeability was 13.38 barrer, along with the $CO_2/CH_4$

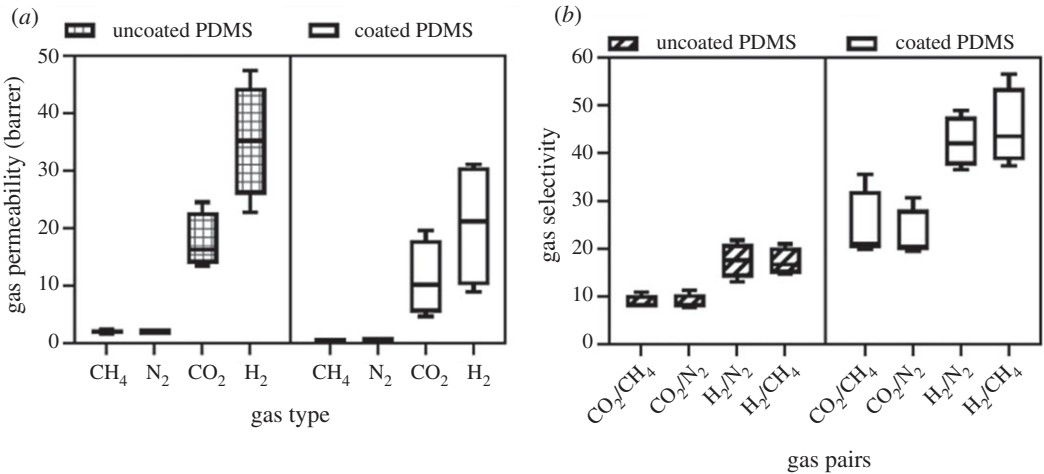

**Figure 10.** The effect of PDMS treatment toward the overall (*a*) permeability and (*b*) selectivity performance of the MMMs.

and $CO_2/N_2$ selectivity of 8.64 and 7.73, respectively. It is suspected that minor surface defect was formed during the membrane potting owing to mishandled treatment. An attempt to fix the defect was by the PMDS coating (3 wt% PDMS in 97% *n*-hexane) on the potted membrane. Then the actual result of the membrane performance was obtained. It is already known that employing low concentration of PDMS could aid the minor defect formation on the membrane surface and result in actual gas separation properties of polymer [5,60].

For $CO_2$ separation, the neat membrane exhibits $CO_2$, $N_2$ and $CH_4$ permeability of 4.68, 0.22 and 0.21 barrer, respectively. The $CO_2/CH_4$ and $CO_2/N_2$ selectivity of 21.89 and 20.93, respectively, is consistent with the previously reported data [39,61]. It was noticed that the permeability of all measured gas improved after the incorporation of ZTC. It would be considered that at 0.5 wt% ZTC loading, the $CO_2$ permeability improved to 7.06 barrer, which also improved the slow gas permeability of $N_2$ and $CH_2$ by 1.6 and 1.7, respectively. Furthermore, the MMMs showing the best performance at 1 wt% loading with $CO_2$ improved from 4.68 into 19.57 barrer along with the $CO_2/CH_4$ and $CO_2/N_2$ selectivity from 21.89 to 35.54 and 20.93 to 30.68, respectively. At higher loading of 1.5 wt%, there was a performance drop observed on the MMMs, even having lower selectivity when compared with the neat membrane.

For $H_2$ separation, the MMMs selectivity performance reduction was observed in the low loading. Permeability-wise, all the ZTC filled membranes exhibit $H_2$ permeability boost when compared with the neat membrane. The $H_2$ permeability of neat, 0.5, 1 and 1.5% ZTC loading was 8.95, 13.15, 31.09 and 29.10 barrer, respectively. This result showed similar behaviour to the $CO_2$ separation performance, which decreased selectivity at low loading. Generally, performance drop was observed at high filler loading owing to the aggregate formation resulting in the void formation. In this study, the aggregate was indeed observed on the high loading as can be seen in the SEM data. However, from the AFM result, the surface defect was not observed, thus the performance drop might be attributed to the filler properties toward permeate gas and the filler position on the membrane surface, which will be discussed later.

The PDMS treatment toward the overall MMMs membrane gas separation performance is illustrated in figure 10. It can be seen the PDMS treatment was slightly reducing the gas permeability through all membrane samples. The reduced permeability was accompanied with significant selectivity boost by two- to three-fold. This was owing to the reduction of pinholes on the membrane surface by the PDMS coating. PDMS is a rubbery material with low selectivity and high permeability that is composed of high flexibility of silane group [62]. Our previous result observed that PDMS has very good compatibility with P84 by forming uniform layer on the membrane surface [5].

As can be seen in figure 11, the gas permeability decreases as the kinetic diameter of the gas increases, typical trend for the molecular sieving ability [11]. Strong molecular sieving characteristic was observed in the P84/ZTC 1 with clear permeability drop as the molecular size of the gas increases. This indicates the ZTC particle at this loading mostly controls the gas transport through the membrane. As discussed previously, ZTC has two pore characteristics of micropore in the inner and mesoporous region on the outer part of the ZTC body. The mesoporous region was responsible to accelerate the molecule

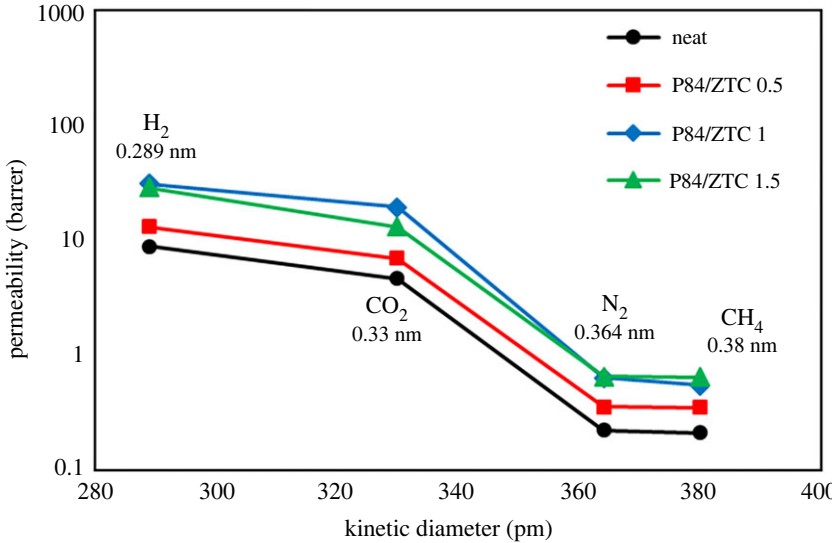

**Figure 11.** Gas permeability versus kinetic diameter of MMMs.

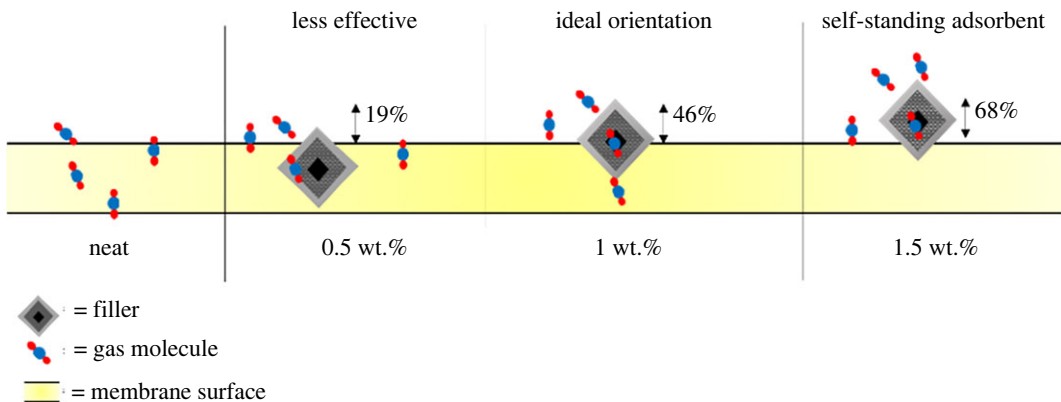

**Figure 12.** Proposed separation process on each ZTC loading.

diffusion to the membrane, while the microporous region controls the selectivity. Moreover, the separation factors of $H_2/N_2$ and $H_2/CH_4$ as well as $CO_2/CH_4$ and $CO_2/N_2$ were quite decent. This indicates that P84/ZTC 1 has potential for gas separation process, even for the case of $N_2/CH_4$ that has selectivity of 1.16. It is slightly passing the expected Knudsen separation (1.07).

Previously, it was assumed that the difference in $CO_2$ and $H_2$ separation behaviour on this membrane was owing to filler characteristic toward permeating gas and filler position on the membrane surface. First of all, as described by Nishihara *et al.* [36], ZTC was already widely used in the gas adsorption process involving $H_2$ [38,63], $CO_2$ [42,43] and $CH_4$ [36,64]. It was revealed that at ambient temperature and pressure, ZTC is more active toward $CO_2$ when compared with the $H_2$ and $CH_4$ [36]. Based on that, the difference behaviour on the $CO_2$ and $H_2$ separation in the MMMs might be attributed to the partial adsorption process by ZTC particle. The partial adsorption would only be possible when the filler has proportional position as self-standing adsorbent on the membrane surface. Based on the AFM study, filler position on the membrane surface is proposed (figure 12). On the neat membrane, the gas transport was solely controlled by the solution-diffusion mechanism [65,66]. Moving into 0.5 wt% loading, it was observed that only a small portion of ZTC particle was exposed. At this position, the filler gives lesser impact toward the gas transport process and the gas transport still dominated with the solution diffusion. Consequently, slight gas separation performance improvement was observed. Moreover, for the case of $H_2$ separation, the selectivity of $H_2/CH_4$ and $H_2/N_2$ was lower when compared with the neat membrane. On this loading composition, the filler just acts as matrix expander and increases the free volume of the membrane, as discussed previously in XRD result. Thus, only permeability improvement was observed. The ideal position was at 1 wt%

loading of ZTC. At this point, the ZTC plays an important role in the gas transport, by having almost 50% portion being exposed. The mesoporous site provides acceleration toward the gas diffusivity, while the micropore controls the selectivity. Moreover, the previous section demonstrated that the permeability was typical to the ability of molecular sieving. Consequently, significant boost of the membrane gas separation performance was observed. At later loading composition, aggregate was observed and only a small part of ZTC particle was dipped on the membrane surface. On this position, it incorporated a self-standing adsorbent on the membrane surface that resulted in the partial adsorption indicated by the permeability drop when compared with the 1 wt% of ZTC loading. Our previous study revealed that ZTC was able to adsorb decent $CO_2$ with fast kinetics rate, thus high $CO_2$-related separation process drop was observed [37]. In contrast for the $H_2$ separation, the performance was less affected because ZTC is a less active $H_2$ adsorbent at room temperature and low pressure [36].

## 3.4. Mixed gas permeation test

Mixed gas separation performance of the uncoated MMMs was examined using two different gas feeds including equimolar $CO_2/CH_4$ and $H_2/CH_4$ gas pair. The mixed gas experiments were performed at room temperature and feed pressure of 2 bar (electronic supplementary material, table S1). The performance trend of the mixed gas separation still follows the single gas result. However, significant reduction in the permeability and selectivity was observed. For the neat membrane, performance drop in competitive environment may be ascribed to various factors such as competitive sorption, concentration polarization, gas phase non-ideality and polymer plasticization [47,67]. In the presence of $CH_4$, the sorption of $CO_2$ in glassy polymers hindered the diffusion pathways for the smaller interactive molecules such as $CO_2$ and $H_2$. For the case of $H_2/CH_4$, the sorption was preferable for the $CH_4$ owing to higher critical temperature (190.55 K) when compared with $H_2$ (33.20 K) and consequently led into $H_2$ permeability drop. For the $CO_2/CH_4$ case, the presence of $CH_4$ decreased the $CO_2$ solubility coefficient which resulted in the $CO_2$ permeability drops which were also observed previously [47,67]. The rise of the $CH_4$ adsorption/solubility in pure P84 was also observed for MMMs, but the increment was less. It can be stated that the $CH_4$ solubility/sorption decreased in the presence of ZTC particle. However, owing to high microporous content in the inside of ZTC particle, pore blockage by large gas molecule of $CH_4$ would result in the delayed diffusion on the smaller gas molecule such as $CO_2$ and $H_2$. On the case of $H_2$ gas molecule, the reduction in permeability was a lot more when compared with the $CO_2$ gas molecule. For instance, in the P84/ZTC 1.5 membrane, the $H_2$ permeability and $H_2/CH_4$ selectivity drop was 39.06 and 23.54%, respectively. This strengthens the assumption that ZTC is a highly $CO_2$-active material.

The membrane performance was compared with the other reported literature (see our dataset in data availability) with respect to the Robeson upper bound curve. As can be seen in figure 13, all the MMMs still suffered under the 2008 Robeson upper bound. However, all the studied MMMs exhibit outstanding performance when compared with other literature data. The membrane loaded with 1 wt% of ZTC showing the best performance in all studied gas separation performance. Moreover, this membrane lies in the 1991 upper bound for $CO_2/CH_4$ separation. At this loading, the filler was in perfect position to provide alternative pathway of gas molecule. The mesopores give an improvement toward the $CO_2$ and $H_2$ permeability to 19.57 and 31.09 barrer, respectively, while the $CO_2/N_2$, $CO_2/CH_4$, $H_2/N_2$ and $H_2/CH_4$ selectivity was greatly improved from 21.93 to 30.68, 21.89 to 35.54, 39.98 to 48.86 and 41.81 to 56.53, respectively. On contrast, the MMMs performance was not significantly improved, or even decreased at earlier or later loading composition. For mixed gas test, the separation performance reduction was still acceptable as the result was still greater than some of the literature and lies before the single gas performance for uncoated membrane. Overall, it can be stated that the addition of ZTC offered an overall better gas separation performance when compared with the neat membrane, which was indicated by the closer position toward the Robeson upper bound. Moreover, the PDMS coating was able to improve the membrane performance by revealing the true nature of P84 gas separation properties. A special remark was put on the $CO_2/CH_4$ separation. Unlike the graphene sheets material that reduce the membrane permeability after the filler incorporation, the addition of ZTC not only improved the selectivity but also increased the permeability owing to its accessible pores. When compared with other two-dimensional graphene-based filler, the addition of ZTC as three-dimensional graphene material showed decent performance in improving the membrane separation. Even though the membranes present a selectivity/permeability trade-off that is below the Robeson upper bound, these membranes have great potential to be further explored in gas separation application since it has superior performance when compared with the other membrane reported

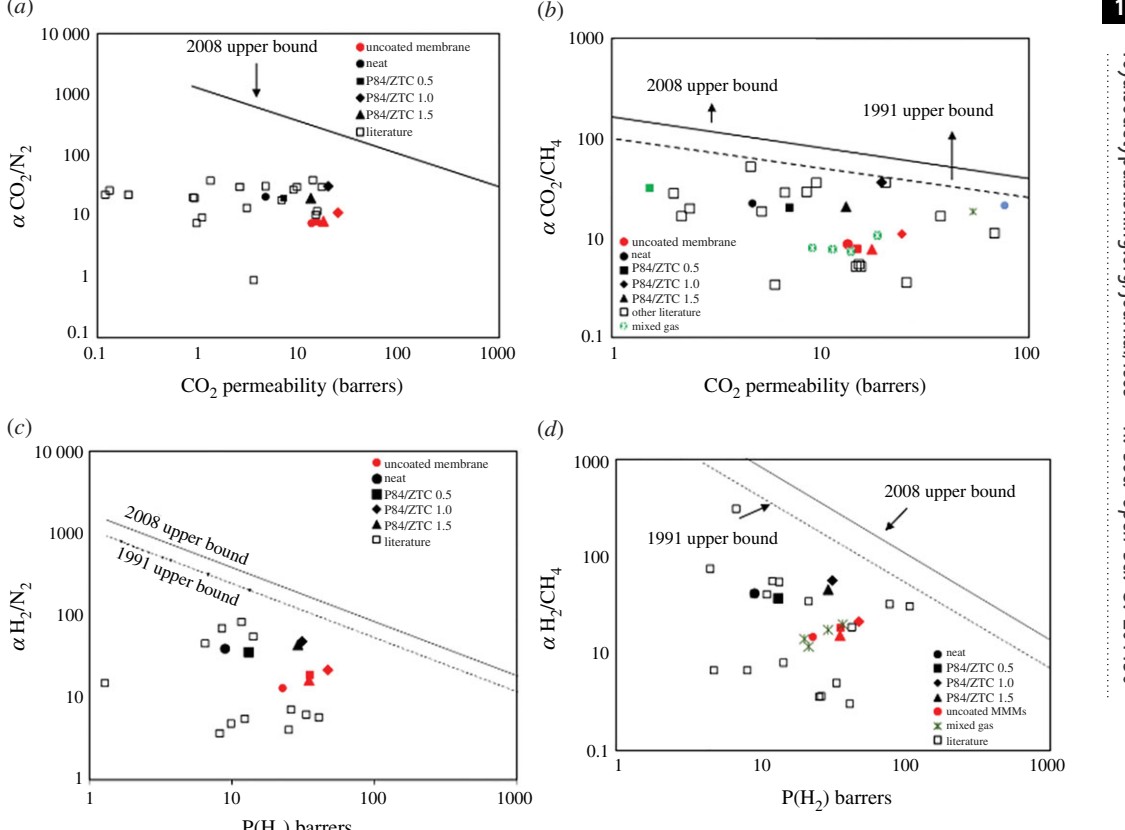

**Figure 13.** The MMMs separation performance for (*a*) $CO_2/N_2$, (*b*) $CO_2/CH_4$ (rectangle 34, asterisk 35, circle 68) (*c*) $H_2/N_2$ and (*d*) $H_2/CH_4$ when compared with other literature data [11,39,61,68–100] with respect of Robeson upper bound curve.

previously. The membrane performance can be furthered improved by advanced treatment such as converting it into carbon membrane. Overall, the unique structure of ZTC that has micro and mesoporous structure was able to improve the membrane performance significantly even though at low loading, which means this membrane can be produced at lower cost.

## 4. Conclusion

This research has conducted the effect of ZTC loading towards the gas separation performance of P84 co-polyimide membrane. As a new filler in this field, ZTC showed a promising performance to enhance the membrane separation performance. The mesoporous and microporous site in the ZTC play an important role in controlling the gas transport through the membrane. Moreover, the filler position on the membrane surface greatly affects the performance outcome of MMMs. At 0.5 wt%, ZTC loading with the filler exposure of 19%, the gas separation improvement was not significant. The $CO_2$ and $H_2$ only improved from 4.68 to 7.06 barrer and from 8.95 to 31.09 barrer, respectively. The optimal loading composition was at 1 wt% with the $CO_2$ and $H_2$ permeability improvement to 19.57 and 31.09 barrer, respectively. Furthermore, at this loading the $CO_2/N_2$, $CO_2/CH_4$, $H_2/N_2$ and $H_2/CH_4$ selectivity was greatly improved from 21.93 to 30.68, 21.89 to 35.54, 39.98 to 48.86 and 41.81 to 56.53, respectively. A partial adsorption process was observed in the 1.5 wt% of ZTC loading which was indicated by the significant gas permeability drop in the binary gas mixture. Overall, the addition of ZTC not only improved the gas separation performance of the P84 but also improved the thermal resistance of the membrane. The MMMs showing potential performance in the gas separation process and the performance can be further improved in the form of carbon membrane.

Data accessibility. Our dataset are deposited in Dryad Digital Repository: https://dx.doi.org/10.5061/dryad.1zcrjdfq0 [101].

Authors' contributions. T.G. carried out the laboratory work, participated in data analysis, carried out characterization, T.G., N.W., H.F., W.N.W.S., participated in the design of the study and drafted the manuscript; R.L., J.M., S.S. carried out the statistical analyses; T.G., R.L. collected field data; N.W., A.F.I., W.N.W.S. conceived of the study, designed the study, coordinated the study and T.G., H.F. helped draft the manuscript. All authors gave final approval for publication.
Competing interests. We declare we have no competing interests.
Funding. We received no funding for this study.
Acknowledgement. The authors would like to appreciate the research funding provided by Institut Teknologi Sepuluh Nopember, under the 'Penelitian Doktor Baru', contract no. 866/PKS/ITS/2020.

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
