## [Peer Review File · Royal Society Open Science]

Review History

RSOS-201150.R0 (Original submission)

Review form: Reviewer 1

Is the manuscript scientifically sound in its present form?

Yes

Are the interpretations and conclusions justified by the results?

Yes

Is the language acceptable?

Yes

Do you have any ethical concerns with this paper?

Yes

Have you any concerns about statistical analyses in this paper?

Yes

Recommendation?

Major revision is needed (please make suggestions in comments)

Comments to the Author(s)

This manuscript deals with the synthesis and gas permeation analysis of mixed matrix membranes, obtained combining zeolite templated carbon (ZTC) as a filler into P84 hollow fiber membranes. The ZTC loading enhances the separation performance of the P84 co-polyimide membranes for CO₂/N₂, CO₂/CH₄, H₂/N₂ and H₂/CH₄; the optimal loading amount is 1 wt%. The topic is interesting, but the manuscript requires some major revisions.

My comments are reported below.

- Most of the discussion, tables and figures are focused on the single gas behavior. Nevertheless, the separation performance of a membrane can be better appreciated in mixture conditions. The authors should add and comment some figures/tables of the mixture results;
- I have not found Table S1 in the supplementary file;
- Tables 3 and 4 show permeability and selectivity at different ZTC loading. It seems that the effect of loading is similar for CO₂ and H₂. Nevertheless, the authors referred to an opposite behavior of H₂ compared to CO₂ (page 8, lines 18-20). Could the authors clarify this point in the manuscript?
- Figure 13 shows that the synthesized membranes present a selectivity/permeability trade off that is below the Robeson upper bound. The authors should better explain in the manuscript the advantages of using these membranes.
- Reference source not found at page 7.

Review form: Reviewer 2

Is the manuscript scientifically sound in its present form?

Yes

Are the interpretations and conclusions justified by the results?

Yes

Is the language acceptable?

No

Do you have any ethical concerns with this paper?

No

Have you any concerns about statistical analyses in this paper?

No

Recommendation?

Major revision is needed (please make suggestions in comments)

Comments to the Author(s)

In your manuscript, zeolite templated carbon (ZTC), a structurally unique of 3D graphene, was prepared by the impregnation of sucrose into zeolite-Y pores and utilized as membrane filler in hollow fiber BTDA-TDI/MDI (P84) co-polyimide membrane for gas separation. Moreover, the effect of the filler position on the membrane surface was examined as well. The preparation and fabrication of ZTC/polymeric membrane composite was conducted at a series of ZTC/P84 composite membranes loaded with ZTC contents (0-1.5 wt%) by the dry/wet spin method. Detailed characterization of morphology (SEM), thermal stability (TGA), crystalline structure

(XRD), topology (AFM) and functional group (FTIR) were conducted to understand the properties of nanocomposite polymeric membranes. It has a certain significance to improve the performance of P84 hollow fiber membrane for gas separation. But there were some details that are problematic here. I believe the paper may be accepted for publication after minor revision

1. First of all, there are some grammatical errors in your manuscript, which are listed below:

1) In the page 2(L34 to35), there seems to be a lack of prepositions after the word "fast". Please consider adding a preposition.

2) In the page 2(L49 to51), "Graphene is a well know 2D carbon material composed of a single layer of carbon atom with a potential of CO₂ adsorption". The word "know" is not suitable for this sentence. Please consider using other words.

3) In the page 2(L51 to54), "Moreover, it has excellent thermal and mechanical stabilities and it is very beneficial to be embedded into polymer matrix to improve its physical properties." It seems that "stabilities" is an uncountable noun with no plural form. Please consider replacing this word.

4) In the page 3(L13), it is recommended to replace "utilize" with "utilizing".

5) In the page 5(L31), it is recommended to replace "with" with "by".

6) In the page 5(L41), it is recommended to delete the "was" before "lies".

7) In the page 5(L43), the preposition in seems to be wrong in this sentence. Please consider replacing this word.

8) In the page 5(L23), it is recommended to add the article "a" before "higher".

9) In the page 5(L29), before "peak", the adjective "appeared" seems redundant. Please consider removing it.

10) In the page 6(L40), it is recommended to replace "indicate" with "indicates".

11) In the page 7(L17), the countable noun "attachment" needs to use its plural form after a plural modifier. Please consider using plural nouns or variable plural modifiers as singular modifiers.

12) In the page 7(L20), it seems that the subject is inconsistent with the predicate verb "was". Please consider changing the verb form.

13) In the page 7(L30), the noun "increase" seems to be used incorrectly. Please consider replacing it.

14) In the page 7(L31), the verb "showing" seems to be wrong. Please consider replacing this verb.

2. In the page 7 (L 55 to 59), part of the content of the manuscript seems to be missing, please complete it.

3. In the page 8(L14 to15), part of the content of the manuscript has a display error.

4. The application of carbon nanomaterials as membrane filler including carbon nanotube, graphene and composites has been reported, which need to be reviewed completely in introduction, such as Journal of Membrane Science, 2013; 448:81-92; Journal of Membrane Science, 2016, 520:281-293; ACS Applied Materials & Interfaces, 2016, 8:18418-18429; Applied Surface Science, 2018,428: 990-999

5. In the page 9(L24 to 28), "However, from the AFM result, the surface defect was not observed, thus the performance drop might be attributed to the filler properties toward permeate gas and the filler orientation on the membrane surface, which will be discussed later.", The word "orientation" in this sentence is not very accurate. The second half of the manuscript only discusses the position of the filler on the surface of the film without considering its orientation.

Decision letter (RSOS-201150.R0)

Dear Dr Gunawan:

Title: The Utilization of Micro-Mesoporous Carbon-Based Filler in the P84 Hollow Fiber Membrane for Gas Separation
Manuscript ID: RSOS-201150

The editor assigned to your manuscript has now received comments from reviewers. I apologise that this has taken longer than usual. We would like you to revise your paper in accordance with the referee and Subject Editor suggestions which can be found below (not including confidential reports to the Editor). Please note this decision does not guarantee eventual acceptance.

Please submit your revised paper before 11-Dec-2020. Please note that the revision deadline will expire at 00.00am on this date. If we do not hear from you within this time then it will be assumed that the paper has been withdrawn. In exceptional circumstances, extensions may be possible if agreed with the Editorial Office in advance. We do not allow multiple rounds of revision so we urge you to make every effort to fully address all of the comments at this stage. If deemed necessary by the Editors, your manuscript will be sent back to one or more of the original reviewers for assessment. If the original reviewers are not available we may invite new reviewers.

On behalf of the Subject Editor Professor Anthony Stace and the Associate Editor Dr Dattatray Late.

RSC Associate Editor:
Comments to the Author:
Major Revision needed

RSC Subject Editor:
Comments to the Author:
(There are no comments.)

Reviewers' Comments to Author:
Reviewer: 1

Comments to the Author(s)

This manuscript deals with the synthesis and gas permeation analysis of mixed matrix membranes, obtained combining zeolite templated carbon (ZTC) as a filler into P84 hollow fiber membranes. The ZTC loading enhances the separation performance of the P84 co-polyimide membranes for CO₂/N₂, CO₂/CH₄, H₂/N₂ and H₂/CH₄; the optimal loading amount is 1 wt%. The topic is interesting, but the manuscript requires some major revisions.

My comments are reported below.

- Most of the discussion, tables and figures are focused on the single gas behavior. Nevertheless, the separation performance of a membrane can be better appreciated in mixture conditions. The authors should add and comment some figures/ tables of the mixture results;
- I have not found Table S1 in the supplementary file;
- Tables 3 and 4 show permeability and selectivity at different ZTC loading. It seems that the effect of loading is similar for CO₂ and H₂. Nevertheless, the authors refereed to an opposite behavior of H₂ compared to CO₂ (page 8, lines 18-20). Could the authors clarify this point in the manuscript?
- Figure 13 shows that the synthesized membranes present a selectivity/permeability trade off that is below the Robeson upper bound. The authors should better explain in the manuscript the advantages of using these membranes.
- Reference source not found at page 7.

Reviewer: 2

Comments to the Author(s)

In your manuscript, zeolite templated carbon (ZTC), a structurally unique of 3D graphene, was prepared by the impregnation of sucrose into zeolite-Y pores and utilized as membrane filler in hollow fiber BTDA-TDI/MDI (P84) co-polyimide membrane for gas separation. Moreover, the effect of the filler position on the membrane surface was examined as well. The preparation and fabrication of ZTC/polymeric membrane composite was conducted at a series of ZTC/P84 composite membranes loaded with ZTC contents (0-1.5 wt%) by the dry/wet spin method. Detailed characterization of morphology (SEM), thermal stability (TGA), crystalline structure (XRD), topology (AFM) and functional group (FTIR) were conducted to understand the properties of nanocomposite polymeric membranes. It has a certain significance to improve the

performance of P84 hollow fiber membrane for gas separation. But there were some details that are problematic here. I believe the paper may be accepted for publication after minor revision

1. First of all, there are some grammatical errors in your manuscript, which are listed below:

- 1) In the page 2(L34 to35), there seems to be a lack of prepositions after the word "fast". Please consider adding a preposition.
 - 2) In the page 2(L49 to51), "Graphene is a well know 2D carbon material composed of a single layer of carbon atom with a potential of CO2 adsorption". The word "know" is not suitable for this sentence. Please consider using other words.
 - 3) In the page 2(L51 to54), "Moreover, it has excellent thermal and mechanical stabilities and it is very beneficial to be embedded into polymer matrix to improve its physical properties." It seems that "stabilities" is an uncountable noun with no plural form. Please consider replacing this word.
 - 4) In the page 3(L13), it is recommended to replace "utilize" with "utilizing".
 - 5) In the page 5(L31), it is recommended to replace "with" with "by".
 - 6) In the page 5(L41), it is recommended to delete the "was" before "lies".
 - 7) In the page 5(L43), the preposition in seems to be wrong in this sentence. Please consider replacing this word.
 - 8) In the page 5(L23), it is recommended to add the article "a" before "higher".
 - 9) In the page 5(L29), before "peak", the adjective "appeared" seems redundant. Please consider removing it.
 - 10) In the page 6(L40), it is recommended to replace "indicate" with "indicates".
 - 11) In the page 7(L17), the countable noun "attachment" needs to use its plural form after a plural modifier. Please consider using plural nouns or variable plural modifiers as singular modifiers.
 - 12) In the page 7(L20), it seems that the subject is inconsistent with the predicate verb "was". Please consider changing the verb form.
 - 13) In the page 7(L30), the noun "increase" seems to be used incorrectly. Please consider replacing it.
 - 14) In the page 7(L31), the verb "showing" seems to be wrong. Please consider replacing this verb.
2. In the page 7 (L 55 to 59), part of the content of the manuscript seems to be missing, please complete it.
3. In the page 8(L14 to15), part of the content of the manuscript has a display error.
4. The application of carbon nanomaterials as membrane filler including carbon nanotube, graphene and composites has been reported, which need to be reviewed completely in introduction, such as Journal of Membrane Science, 2013; 448:81-92; Journal of Membrane Science, 2016, 520:281-293; ACS Applied Materials & Interfaces, 2016, 8:18418-18429; Applied Surface Science, 2018,428: 990-999
5. In the page 9(L24 to 28), "However, from the AFM result, the surface defect was not observed, thus the performance drop might be attributed to the filler properties toward permeate gas and the filler orientation on the membrane surface, which will be discussed later.", The word "orientation" in this sentence is not very accurate. The second half of the manuscript only discusses the position of the filler on the surface of the film without considering its orientation.

Author's Response to Decision Letter for (RSOS-201150.R0)

See Appendix A.

Decision letter (RSOS-201150.R1)

This year has been very difficult for everyone, and we want to take the opportunity to thank you for your continued support in 2020.

The Royal Society Open Science editorial office will be closed from the evening of Friday 18 December 2020 until Monday 4 January 2021. We will not be responding during this time. If you have received a deadline within this time period, please contact us as soon as possible to allow us to extend the deadline. If you receive any automated messages during this time asking you to meet a deadline, we offer apologies and invite you to respond after the festive period or during normal working hours.

With our best for a peaceful festive period and New Year, and we look forward to working with you in 2021.

Dear Dr Gunawan:

Title: The Utilization of Micro-Mesoporous Carbon-Based Filler in the P84 Hollow Fiber Membrane for Gas Separation
Manuscript ID: RSOS-201150.R1

It is a pleasure to accept your manuscript in its current form for publication in Royal Society Open Science. The chemistry content of Royal Society Open Science is published in collaboration with the Royal Society of Chemistry.

On behalf of the Subject Editor Professor Anthony Stace and the Associate Editor Dr Dattatray Late.

RSC Associate Editor
Comments to the Author:
Authors have carefully addressed all the issues raised by the reviewer's.

Reviewer(s)' Comments to Author:

Appendix A

Response to the Reviewers' Comments to Author:

Reviewer: 1

Comments to the Author(s)

This manuscript deals with the synthesis and gas permeation analysis of mixed matrix membranes, obtained combining zeolite templated carbon (ZTC) as a filler into P84 hollow fiber membranes. The ZTC loading enhances the separation performance of the P84 co-polyimide membranes for CO₂/N₂, CO₂/CH₄, H₂/N₂ and H₂/CH₄; the optimal loading amount is 1 wt%.

The topic is interesting, but the manuscript requires some major revisions.

My comments are reported below.

- Most of the discussion, tables and figures are focused on the single gas behavior. Nevertheless, the separation performance of a membrane can be better appreciated in mixture conditions. The authors should add and comment some figures/tables of the mixture results;

Answer : We add the mixed gas result in the Robeson curve (Fig 13) and comment for it.

- I have not found Table S1 in the supplementary file;

Answer : Thank you for pointing this out, the data has been added in the supplementary files. Actually the data has been stored in Dryad at <https://doi.org/10.5061/dryad.1zcrjdfq0>.

- Tables 3 and 4 show permeability and selectivity at different ZTC loading. It seems that the effect of loading is similar for CO₂ and H₂. Nevertheless, the authors referred to an opposite behavior of H₂ compared to CO₂ (page 8, lines 18-20). Could the authors clarify this point in the manuscript?

Answer : Thank you for the question, at first we looked at the selectivity of uncoated membrane with ZTC loading of 0.5 wt% and we observe the selectivity reduction on the CO₂/CH₄, and did not observe the same on the H₂/CH₄. So we assume that the membrane has opposite behavior for the stated gas. But after looking at the data once more, we can not agree more with the Reviewer, that the data has similar for both CO₂ and H₂, so we revised our statement becomes "This result showing similar behavior to the CO₂ separation performance...."

- Figure 13 shows that the synthesized membranes present a selectivity/permeability trade off that is below the Robeson upper bound. The authors should better explain in the manuscript the advantages of using these membranes.

Answer : Thank you for the input, we add

"Even though the membranes present a selectivity/permeability trade off that is below the Robeson upper bound, these membranes have great potential to be further explored in gas separation application since it has superior performance as compared to the other membrane reported previously. The membrane performance can be further improved by advanced treatment such as converting it into carbon membrane. Overall, the unique structure of ZTC that has micro and mesoporous structure were able to improve the membrane performance significantly even though at low loading, which makes this membrane can be produced at lower cost." At the end of the result and discussion

- Reference source not found at page 7.

Answer : Thank you, the issue has been fixed accordingly

Reviewer: 2

Comments to the Author(s)

In your manuscript, zeolite templated carbon (ZTC), a structurally unique of 3D graphene, was prepared by the impregnation of sucrose into zeolite-Y pores and utilized as membrane filler in hollow fiber BTDA-TDI/MDI (P84) co-polyimide membrane for gas separation. Moreover, the effect of the filler position on the membrane surface was examined as well. The preparation and fabrication of ZTC/polymeric membrane composite was conducted at a series of ZTC/P84 composite membranes loaded with ZTC contents (0-1.5 wt%) by the dry/wet spin method. Detailed characterization of morphology (SEM), thermal stability (TGA), crystalline structure (XRD), topology (AFM) and functional group (FTIR) were conducted to understand the properties of nanocomposite polymeric membranes. It has a certain significance to improve the performance of P84 hollow fiber membrane for gas separation. But there were some details that are problematic here. I believe the paper may be accepted for publication after minor revision

1. First of all, there are some grammatical errors in your manuscript, which are listed below:

- 1) In the page 2(L34 to35), there seems to be a lack of prepositions after the word "fast". Please consider adding a preposition.
- 2) In the page 2(L49 to51), "Graphene is a well know 2D carbon material composed of a single layer of carbon atom with a potential of CO₂ adsorption". The word "know" is not suitable for this sentence. Please consider using other words.
- 3) In the page 2(L51 to54), "Moreover, it has excellent thermal and mechanical stabilities and it is very beneficial to be embedded into polymer matrix to improve its physical properties." It seems that "stabilities" is an uncountable noun with no plural form. Please consider replacing this word.
- 4) In the page 3(L13), it is recommended to replace "utilize" with "utilizing".
- 5) In the page 5(L31), it is recommended to replace "with" with "by".
- 6) In the page 5(L41), it is recommended to delete the "was" before "lies".
- 7) In the page 5(L43), the preposition in seems to be wrong in this sentence. Please consider replacing this word.
- 8) In the page 5(L23), it is recommended to add the article "a" before "higher".
- 9) In the page 5(L29), before "peak", the adjective "appeared" seems redundant. Please consider removing it.
- 10) In the page 6(L40), it is recommended to replace "indicate" with "indicates".
- 11) In the page 7(L17), the countable noun "attachment" needs to use its plural form after a plural modifier. Please consider using plural nouns or variable plural modifiers as singular modifiers.
- 12) In the page 7(L20), it seems that the subject is inconsistent with the predicate verb "was". Please consider changing the verb form.
- 13) In the page 7(L30), the noun "increase" seems to be used incorrectly. Please consider replacing it.
- 14) In the page 7(L31), the verb "showing" seems to be wrong. Please consider replacing this verb.

Answer : All these issues has been revised accordingly and marked with "yellow" text highlight

2. In the page 7 (L 55 to 59), part of the content of the manuscript seems to be missing, please complete it.

Answer : we are not sure which part of missing or this is just the system error, the sentence "more insoluble ZTC tends to form agglomerate and occupied the bottom part of Duran flask due to the increasing of total weight." is connected to. In addition we removed the sentence "With that being said, the membrane will suffer in selectively loss." (L54 to 55)

3. In the page 8(L14 to15), part of the content of the manuscript has a display error.

Answer :The issue has been addressed and changed to "listed in Table 3 and Table 4"

4. The application of carbon nanomaterials as membrane filler including carbon nanotube, graphene and composites has been reported, which need to be reviewed completely in introduction, such as Journal of Membrane Science, 2013; 448:81-92; Journal of Membrane Science, 2016, 520:281-293; ACS Applied Materials & Interfaces, 2016, 8:18418-18429; Applied Surface Science, 2018,428: 990-999

Answer : This has been added in the introduction "...silica^{18,19}, zeolites^{20,21}, MOF²², graphene²³⁻²⁶"

5. In the page 9(L24 to 28), "However, from the AFM result, the surface defect was not observed, thus the performance drop might be attributed to the filler properties toward permeate gas and the filler orientation on the membrane surface, which will be discussed later.", The word "orientation" in this sentence is not very accurate. The second half of the manuscript only discusses the position of the filler on the surface of the film without considering its orientation.

Answer : Thanks for your input, we addressed this issue by changing the word "orientation" into "position".